# Information Security Applications in Smart Cities: A Bibliometric Analysis of Emerging Research

Thiago Poleto [1,*] , Thyago Celso Cavalcante Nepomuceno [2] , Victor Diogho Heuer de Carvalho [3,*] , Ligiane Cristina Braga de Oliveira Friaes [1] , Rodrigo Cleiton Paiva de Oliveira [1] and Ciro José Jardim Figueiredo [4]

[1] Department of Business Administration, Institute for Applied Social Sciences, Federal University of Pará, Belém 66075-110, Brazil; ligianebraga@ufpa.br (L.C.B.d.O.F.); ro.wright09@gmail.com (R.C.P.d.O.)
[2] Department of Statistics, Center for Exact and Natural Sciences, Federal University of Pernambuco, Recife 50670-901, Brazil; thyago.nepomuceno@ufpe.br
[3] Technologies Axis, Campus do Sertão, Federal University of Alagoas, Delmiro Gouveia 57480-000, Brazil
[4] Department of Engineering, Campus Angicos, Federal Rural University of Semi-Arid, Angicos 59515-000, Brazil; ciro.figueiredo@ufersa.edu.br
* Correspondence: thiagopoleto@ufpa.br (T.P.); victor.carvalho@delmiro.ufal.br (V.D.H.d.C.)

**Abstract:** This paper aims to analyze the intellectual structure and research fronts in application information security in smart cities to identify research boundaries, trends, and new opportunities in the area. It applies bibliometric analyses to identify the main authors and their influences on information security and the smart city area. Moreover, this analysis focuses on journals indexed in Scopus databases. The results indicate that there is an opportunity for further advances in the adoption of information security policies in government institutions. Moreover, the production indicators presented herein are useful for the planning and implementation of information security policies and the knowledge of the scientific community about smart cities. The bibliometric analysis provides support for the visualization of the leading research technical collaboration networks among authors, co-authors, countries, and research areas. The methodology offers a broader view of the application information security in smart city areas and makes it possible to assist new research that may contribute to further advances. The smart cities topic has been receiving much attention in recent years, but to the best of our knowledge, there is no research on reporting new possibilities for advances. Therefore, this article may contribute to an emerging body of literature that explores the nature of application information security and smart cities research productivity to assist researchers in better understanding the current emerging of the area.

**Keywords:** information security; smart city; technical collaborations networks; applications; bibliometric analysis



## 1. Introduction

The advancement of solutions and tools focused on information security for smart cities is gaining prominence worldwide [1–7]. Furthermore, there has been a noticeable increase in the production of large volumes of data, agility in information exchange, data analysis (Data Science), development of smart cities, and connectivity between various devices. These continuous interactions with internet-focused solutions (Internet of Things—IoT) must be conducted in compliance with regulations [8–10]. However, they concurrently introduce profound challenges, especially in terms of data governance, and there is a growing emphasis on safeguarding the integrity, confidentiality, and availability of data as it is generated, processed, and exchanged across diverse entities, spanning from private organizations to public sectors and the general populace [11–14].

As public services gravitate towards interconnected digital ecosystems, we can identify significant potential benefits, such as streamlined operations and bolstered resilience in

critical infrastructures. Nonetheless, for metropolises and regions striving to transition into the smart city paradigm, it is imperative to not only meticulously assess but also proactively mitigate the inherent cybersecurity risks stemming from such integration [13–21]. While no technology solution can guarantee complete security, communities need to implement smart city technologies while considering the need to balance efficiency, innovation, and cybersecurity [20,22–25].

This context demands promoting privacy protections, national security, and the secure operation of infrastructure systems. Cities should tailor best practices to their specific cybersecurity requirements, ensuring the protection of citizens' private data as well as the security of sensitive government and business information [20,24,26,27]. By promoting protection through proper guidelines, communities can strive to create a safe and secure environment while embracing the benefits of technological advancements [28].

In recent years, organizations have turned their attention to the increased risks that the lack of information security causes in the evolution and survival of businesses, mainly due to the large offer of technological devices and the growing access and dissemination of data and information [29–32]. The lack of information security evidence many losses for the different business stakeholders, especially when it negatively impacts the trust of customers and suppliers, the efficiency of services, the availability of operations, the credibility of the business, and the image of the company [33]. In this sense, organizations have adopted strategies to prevent the occurrence of security flaws caused by Denial of Service Attacks (DoS), hacking, malware, phishing, spoofing, ransomware, spamming, and other types of cyberattacks [31,34–37]. Strategies, in general, are adopted to protect the business performance and maintain operational efficiency at competitive levels [38]. Thus, excellence in the cybersecurity process is essential to ensure the integrity, availability, and confidentiality of business data and information [39,40].

The discussion over the importance of information security has been highlighted in recent literature. The advancement of research in the area has considered aspects from risk assessment to recovery and resilience of cybersecurity [41,42]. On many occasions, Information Technology (IT) managers seek to analyze solutions to conduct operational strategies aimed at protecting business [43]. In recent years, although many researchers [44–49] have presented approaches to the importance, investment, and contribution of cybersecurity to organizations, society, and government, there is still a gap in the current literature: there are no studies that analyze the most influential works in the area of cybersecurity with an integrated view.

In recent studies on smart cities, there is a growing interest in integrating innovative technologies to optimize urban management and improve the quality of life for citizens. However, upon reviewing the existing literature, a gap is identified in the systematic review related to information security applications in this context. While many studies address the benefits and potential implementations of these technologies, few delve deeply into specific solutions to ensure data protection and user privacy. Given the critical importance of information security in highly connected environments, such as smart cities, this gap presents an opportunity for researchers and IT professionals to delve deeper and contribute with insights and robust solutions to this emerging challenge.

One of the premises for understanding the application of information security in smart city research activities is to analyze its manifestation in the form of scientific production. In this sense, this paper aims to perform a bibliometric analysis to deepen knowledge of new applications of information security in smart cities to identify the main groups of researchers working collaboratively in the area. Moreover, this study provides a summary of research patterns based on an institutional network to present a better understanding of research advances and the latest content about information security in smart cities published in journals during the period from 2015 to 2023. The relevant articles were retrieved from the Scopus database.

The bibliometric analysis allows the visualization of the technical quality and impact of research, as well as grouping authors and co-authors, identifying the relationship

between studies through keywords and number of citations, and displaying intellectual contributions from research fields, among other analyses. In addition, solutions and review of smart cities opens many opportunities and scopes for open research.

This paper is structured as follows: Section 2 presents a theoretical reference with related works about smart cities and information security research; Section 3 is devoted to Materials and Methods; Section 4 presents the Findings and Discussion; Section 5 contributes to the theory and presents practical implications; the conclusion, limitations, and further research are provided in Section 6.

## 2. Smart Cities and Information Security

Before starting a discussion about papers that have reviewed the literature on smart cities, it is essential to address some concepts. A smart city is understood as an urban area where electronic sensor technology is used to collect data from devices as well as assets and citizens for analysis and processing of the data to manage and monitor public infrastructures [50,51]. Smart cities are characterized by the following characteristics in terms of digitalization: Internet of Things (IoT), Big Data, and Cloud Services to promote integration [52,53].

At the heart of a smart city lies a tapestry of devices interconnected via wireless networks, often operating on open network protocols or APIs [54–56]. These elements, by their very design, can be susceptible to breaches, even by the smallest snippets of malicious code [57–59]. Consequently, information security shifts beyond the individual user's realm and emerges as a communal imperative within the smart city landscape [60–62]. Moreover, the escalating intricacy of the system's network infrastructures, magnified by digital communication, interconnected devices, and diverse network architectures, inevitably poses heightened security challenges [1,63–67].

The consequences of successful cyberattacks against smart cities can be severe and wide-ranging. They may include disruptions to essential infrastructure services, substantial financial losses, exposure of citizens' private data, erosion of trust in smart systems, and even physical harm or loss of life due to impacts on physical infrastructure. According to Shin et al. [68], global spending on cybersecurity hardware, software, and services has significantly grown in the past few years, and the annual cybersecurity investment averages USD 1 billion by some financial and tech companies. Cyberattacks are a serious threat to the successful implementation of smart cities-related services. Comprehensive security mechanisms and a security-oriented mindset throughout the entire organization are essential to avert and control this risk.

Table 1 presents the risk domain in information security to smart cities found in the literature, addressing different perspectives on provider and user application of technologies. Upon examination of the table, it is evident that the identified domains encompass topics that resonate with the discussions conducted by experts in the literature, as well as those on Cloud computing, IoT, data interpretation, and smartphone devices. Moreover, the highlighted risks emphasize the imperative need for acquiring deeper insights in advance, specifically in the realm of information security within smart cities, a domain that is growing in significance. Nonetheless, it is worth acknowledging a potential drawback associated with the abundance of published material, which serves as a catalyst for conducting the systematic review presented in this paper to identify guidelines that serve as a contribution to the theme.

The analysis of these works allows us to conclude that information security risk in smart cities is still in the development stage in different devices. Thus, more comprehensive and complete research and analysis of all recent publications in the field of information security is necessary and still lacking. In this sense, a bibliometric study is a valuable tool to present the interrelationships of researchers, their contributions, and the gaps to be worked on.

**Table 1.** Main detected information security risk domains according to literature.

| Area | Risk Domain | References |
|---|---|---|
| Cloud computing (platform of services over the internet, accessible by people and business companies) | Cloud threats | [69–72] |
| | Custodianship of keys | [73] |
| | Security of data | [60,74–77] |
| | Security attacks | [75,78–85] |
| | Lack of a data privacy policy | [73,77,86–92] |
| Internet of Things (concerning devices that have an internet connection and that can communicate with the network independently of human action). | Attacks on IoT devices | [9,35,83,87,93–96] |
| | Lack of effective access controls | [89,97–104] |
| | Protecting sensitive data | [32,105–107] |
| | Botnet activities | [35,108–110] |
| | Privileged user access | [89,99,111] |
| Data interpretation (essentially the representation of complex data and understand trends and follow patterns) | Security reports | [112–114] |
| | Discover sensitive data | [115–118] |
| | Errors and inconsistency Decision | [119–121] |
| | Privacy violations | [122–126] |
| Smartphones (smart communication mobile devices) | Security of data | [127–130] |
| | Smartphone threats | [131,132] |
| | Protecting sensitive data | [133] |
| | Lack of privacy of stakeholders | [134,135] |

*Related Reviews*

The literature on topics associated with information security, cybersecurity, and smart cities contains some systematic literature reviews with very interesting content to assist researchers and practitioners in their definitions in favor of new research and related practical developments. The swift progress of artificial intelligence and data-driven technologies has opened new avenues for tackling intricate socioeconomic issues in the modern world through the utilization of diverse datasets and the application of advanced analytical techniques, fostering inclusive development and sustainable growth in smart cities [136].

The topic of cybersecurity has been a growing concern in scientific literature that extends and is interlinked with many social issues. In the comprehensive review of applications in public security by de Carvalho and Costa [137] spanning materials published between 2014 and the first half of 2021 across significant bibliographic databases like Scopus, Web of Science, IEEE Xplore, and ACM Digital Library, the authors highlight the adaptive techniques and mining techniques to enhance pirate software detection and other security-related concerns.

Following, we present a set of seven systematic reviews related to the one presented in this document, retrieved from the Scopus database. This set was selected based on its impact, measured based on the number of citations.

Habibzadeh et al. [40] developed a survey that provides an overview of both the theoretical and practical challenges and opportunities, considering not only their technical dimensions but also addressing policy and governance concerns. Their study underscores the need for collaborative efforts among different stakeholders to achieve sustainable and secure smart city ecosystems. It offers a comprehensive examination, discussing security and safety implications for critical infrastructures and the resulting policy considerations at various levels. It also assesses privacy and security vulnerabilities inherent in smart city architecture, along with a focus on common smart city applications.

The survey by Sanchez et al. [138] explored the recent advancements in the field of device behavior fingerprinting, examining its applications, sources of behavioral data, and

the techniques employed for processing and assessment. The reliability and performance of emerging environments such as smart cities, Industry 4.0, and crowdsensing depend on the proper functioning of fingerprint devices. This entails a comprehensive grasp of the capabilities of these devices, including sensors and actuators, and the capability to identify potential irregularities arising from cyberattacks, system failures, or misconfigurations.

The survey by Jimada-Ojuolape and Teh [139] provides a comprehensive review of research that extends beyond assessing reliability at the component level and takes into consideration the influence of Information and communication technology integrations on the overall system reliability. The study presents some recommendations based on the literature, which are based on either the adequacy aspect or the security aspect of reliability. It also presents some technological challenges to the reliability of smart grids, going from Infrastructure failures due to cyber–physical interdependencies, passing through environmental aspects, such as the weather conditions, reaching combatting cybersecurity vulnerabilities, such as intrusions/infiltrations.

Kim et al. [140] conducted a systematic and comprehensive investigation of autonomous vehicles by analyzing 151 papers published between 2008 and 2019. They categorized autonomous attacks into three main groups: those targeting the autonomous control system, components of autonomous driving systems, and vehicle-to-everything communications. Protection against these attacks was categorized into security architecture, intrusion detection, and anomaly detection. With advancements in big data and communication technologies, there is a gradual evolution of techniques that employ artificial intelligence and machine learning for anomaly detection. Their survey suggests that future research in autonomous attacks and defenses should be closely integrated with artificial intelligence, as it constitutes a critical component of smart cities.

Alotaibi and Barnawi [141] present a thorough examination of security considerations for massive Internet of Things (IoT) within the context of 6G networks, with a particular focus on Intrusion Detection Systems (IDS). The authors claim this is the inaugural survey to encompass the amalgamation of Machine Learning (ML), Deep Learning (DL), and essential networking technologies that underpin the forthcoming 6G infrastructure for securing massive IoT. As future trends for 6G, they highlight self-adaptive intrusion detection systems, the use of federated learning, self-supervised learning, quantum machine learning, explainable artificial intelligence, transfer learning, and big data technologies, supporting the development of intelligent protection platforms.

Raimundo and Rosário [142] examined the prevailing literature trends concerning the opportunities and threats in Industrial Internet of Things (IIoT) cybersecurity. They have reviewed 70 pivotal articles identified through an extensive survey of the Scopus database, intending to outline the ongoing discourse surrounding IIoT rather than proposing specific technical remedies for network security issues. The study highlighted key themes in the current debate on the involved topics, considering: (i) a cybersecurity axis, observing platforms that may accommodate smart objects, issues related to smart grids in IoT-controlled environments, critical technologies, best practices, policies, and frameworks; (ii) a machine learning axis, to encompass artificial intelligence techniques in cybersecurity; (iii) an IoT axis that considers the use of artificial intelligence combined to physical devices supporting cybersecurity measures for systems protection; (iv) an Industry 4.0 (or IIoT) axis covering industrial applications of IoT and artificial intelligence, also demanding concern about the security of the systems involved; and (v) blockchain and cloud computing axis, representing the decentralized architectures needed to run all the previous concepts plans and technologies.

Yang et al. [143] developed a systematic overview of research related to these technologies, which includes four key components. First, they present a summary of urban sensor concepts and applications. Second, they analyze the progress in multisource heterogeneous urban sensor access technologies, encompassing communication protocols, data transmission formats, access standards, access technologies, and data transmission methods. Third, they review data management technologies for urban sensors, focusing on data cleaning,

data compression, data storage, data indexing, and data querying. Fourth, they address challenges associated with these technologies and propose viable solutions, specifically in the realms of integrating massive Internet of Things (IoT), managing computational load, optimizing energy consumption, and enhancing cybersecurity. Finally, the paper concludes by summarizing their work and hinting at potential future development directions.

## 3. Materials and Method

The bibliometric analysis uses statistical methods to evaluate the evolution of a particular research area. In this sense, it is possible to (i) evaluate the number of publications, the level of quality, the impact, and the contribution of the results; (ii) to carry out a mapping of the scientific activities of the authors; (iii) to understand networks of citations based on the authors; (iv) to obtain a real and detailed visualization of the results and intellectual structures of a scientific domain; (v) to promote the construction of knowledge; (vi) to monitor the evolution of a research field and (vii) to clarify unexplored research topics.

In the past ten years, the advance of cybersecurity research has developed significantly by influential authors in different journals and research areas. The present study consists of a technical and structured analysis of the progress of literature on cybersecurity, with the objectives of presenting collaborations in the editorial production of researchers, highlighting new insights on the role of information security engineering in the world, and stimulating development on future research lines. To direct the research, some questions are posed:

- *Q1—What are the patterns of information security applications found in research on smart cities?*
- *Q2—What are the most demanding areas for information security in smart cities studies?*
- *Q3—What research has the most influence on the application of information security in smart cities?*

To answer these questions, this study adopts a theoretical approach, aiming to understand the state-of-the-art information security and smart cities research fields through bibliometrics and content analysis. Figure 1 shows the research design used in this paper, which consists of five steps.

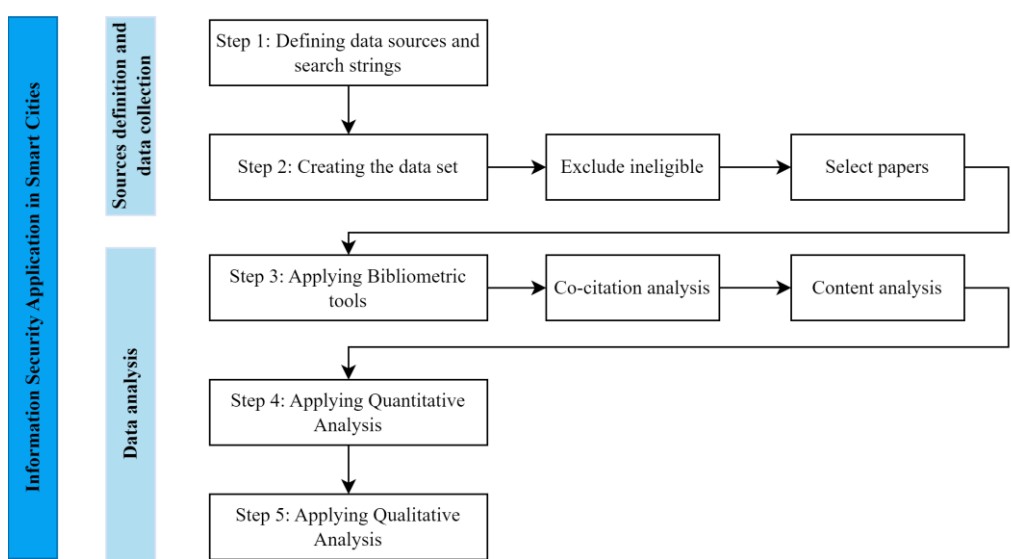

**Figure 1.** Research design.

Step 1 starts with the data sources definition, considering the Scopus database, followed by search string creation. In this study, two combinations of keywords were defined to compose the search string: (I) "information security" and "smart city"; and (II) "cyberattacks" and "smart city". These terms are broad and expand the knowledge about

the different knowledge application areas of the theme. The search was applied to titles, abstracts, and keywords of complete published articles.

In Step 2, the dataset consists of complete articles published in journals indexed in Scopus, ranging from 2015 to 2023. We decided to start searching for published results from 2015 due to the high number of citations from one of the articles of greater relevance to the area, published in the same year.

The work entitled "Cyber security challenges in smart cities: Safety, security and privacy", indicated in the reference list, has obtained 650 citations to date [15]. For this reason, we consider this time interval as the most relevant to collect data. A filter was used to remove articles that emerged from books, categorized. The purpose of using this filter was to focus on the article and conference reviews with significant academic impact and relevance in the research platform. In addition, other categories of publications have also been removed, so the objective is to identify the sectors and fields in which there are one or more surveys and the sectors and methods in which there are no surveys available. The Scopus database was selected due to the broad approach of indexed sources among journals, conferences, and books, increasing the range of data collection for the bibliometrics analyses.

As shown in Figure 2, there is a significant increase in articles on information security and smart cities. The search results returned a total of 1978 articles, including conference papers (55.5%) and journal articles (44.5%).

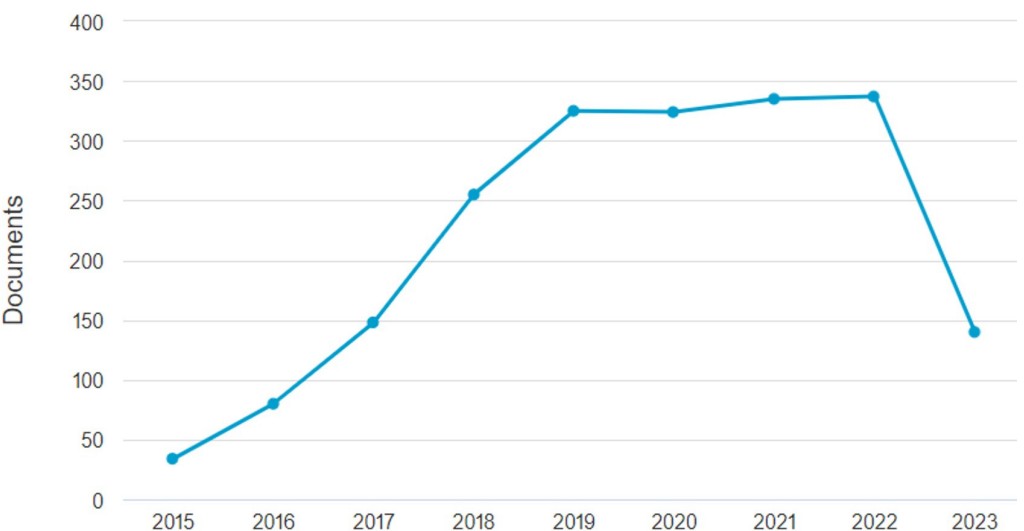

**Figure 2.** Trend line based on the number of publications by year in the field of information security in smart cities.

In Step 3, the VOSViewer software [144], which is a text-mining tool that supports comprehensive and useful compilation of metadata, supporting data generation, and graph visualization, was used as a bibliometric analysis tool.

In Step 4, the quantitative analysis involved the implementation of statistical, network, and content methods through the development of descriptive and cluster analyses comprising information regarding articles, journals, authors, citations, references, and keywords in terms of annual progress in the field of cybersecurity research. The objective was to discover the implications of quantitative results in terms of the historical development of the application of information security and smart cities research field, its patterns, and evolution to answer the three research questions.

Finally, in Step 5, qualitative analysis was used to investigate production indicators (most productive authors, number of publications, types of authorship, area of training), the international authors who constitute the research interface in the area or related areas, and the information security and smart cities community. Also, the analysis of citations and their different relationships contributed to the identification of epistemological, methodological,

and theoretical influences in the domain investigated. From this, through distinctive classifications and thesaurus, the universe of articles analyzed was categorized, which allowed identifying the gaps regarding the study object and contributing to improving the representation schemes on smart cities knowledge.

## 4. Findings and Discussion

The advancement of IT and the emergence and growth of the internet led organizations to adopt new business models based on the potential market focused on creating and using cyberspace information. This business model allows organizations to obtain advantages, but on the other hand, they need to face several problems related to cyberspace security management, which are currently quite prominent.

The first publication in the area is "Cyberspace Security Management," published in 1999 by Chou et al. [145] in the journal of Industrial Management and Data Systems. This first publication evidences the leading causes of Internet security incidents. It starts the discussion about real concerns involving inherent risks, technology weaknesses, policy weaknesses, unauthorized intruders, and legal issues often provoked by players, which affect several business and government organizations in cyberspace. Chou et al. define the users, business sectors, and regulatory agents as leading players that influence the evolution of business and can interfere with principles of cybersecurity, such as confidentiality, integrity, and availability of data and information. The contributions of Chou et al. encourage the development of discussions on potential techniques, methodologies, and investment in IT solutions that address issues related to cybersecurity. As a result, several authors developed studies associated with the area and presented the results of a significant impact on the literature. Therefore, an analytical study of the main trends in the field, discussed in recent years, is suitable.

### 4.1. Identifying the Information Security Applications in Smart Cities Clusters of Research through Bibliographic

To analyze and visualize the knowledge clusters of research on information security applications in smart cities, the graph of relation in Figure 3 was created, considering the authors' groups according to application theme.

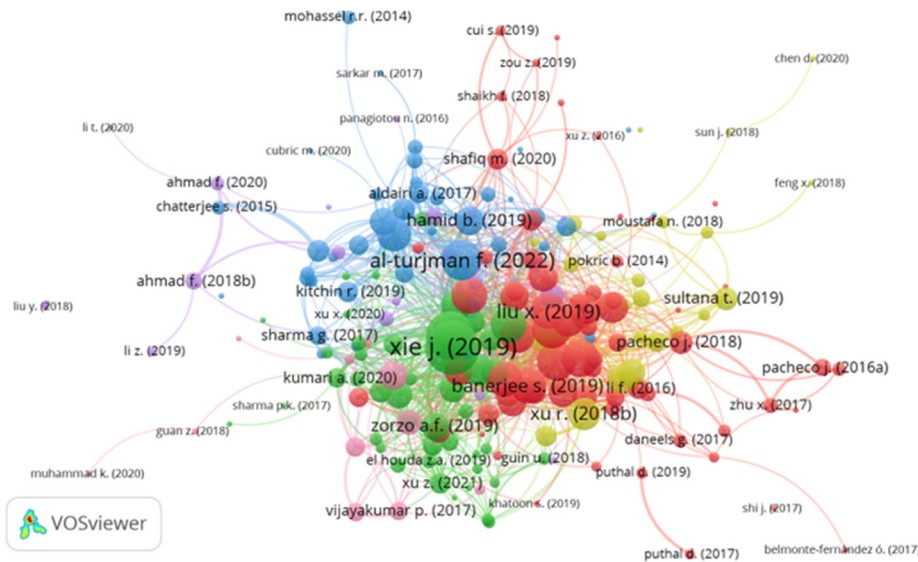

**Figure 3.** Clusters of authors according to applications about information security in smart cities.

The depiction of inter-publication relationships is facilitated by the quantity of links and the spatial proximity of nodes within the visual representation of Figure 3. Each node (circle) on the map corresponds to a publication, and the size within this visualiza-

tion is indicative of the volume of citations received by a respective publication. Proximity in the visualization denotes a stronger correlation, as determined by co-citation patterns, be-tween publications situated closely compared to those positioned at a greater distance [137,146–149]. The linkages between nodes serve to elucidate co-occurrence relationships, with closely associated term clusters forming tightly coupled groups [146,149]. This ap-plication of VOSviewer's co-occurrence analysis emerges as a robust method for constructing conceptual maps, enabling the identification of pivotal ideas and themes within a dataset and facilitating the visual representation of their interconnections in an accessible manner [147,149].

Table 2 details the cluster's compositions, separating them by name (related to the application domain) and listing their sizes as well as the most representative articles.

**Table 2.** Cluster identification with related domain, size, and most representative articles.

| Cluster Number/Color | Cluster Name | Size | Representative Articles |
|---|---|---|---|
| Cluster 1/Red | Smart Power Grid in Smart Cities | 324 | [3,55,71,83,99,111,150–215] |
| Cluster 2/Green | Authentication in Smart Cities | 241 | [22,51,63,85,91,93,94,166,216–272] |
| Cluster 3/Blue | Cyberattacks in Smart Cities | 153 | [1,4,273–293] |
| Cluster 4/Yellow | Security platforms for Smart cities | 121 | [60,294–309] |
| Cluster 5/Pink | Evaluation of threats to cybersecurity | 99 | [6,54,310–324] |
| Cluster 6/Purple | Cybersecurity and society | 78 | [325–334] |

Following, a description of each cluster is provided.

### 4.1.1. Cluster 1 (Red): Smart Power Grid in Smart Cities

One of the applications of information security is related to smart power grid maintenance in smart cities. A smart power grid can offer support to a smart communications grid since society increasingly requires information transfer infrastructure in daily activities [65]. Over the years, utilities have invested in communication networks to improve awareness of the power grid assets and to control, automate, and integrate the service delivery systems. The key point of integrating systems and working in real-time is connectivity. Most of the time, the web facilitates systems integration and benefits society with this support.

On the other hand, the web environment allows targeted attacks and attempts to break into the system. The North American Electric Reliability Corporation [335] highlighted compliance concerns in strengthening essential cybersecurity across the entire power system and emphasized that this requires a series of cybersecurity concerns [87,88,336,337].

For some authors, the smart grid needs to be observed and measured before being controlled and automated [338]. To that end, the automation of the power substation helps utilities add sophisticated protection and control functions while offering more visibility into the performance and integrity of the network infrastructure. Also, it is essential to note that the resilience of physical and electrical networks must also be improved according to the flow of information, as critical operations can cause failures or can be combined with physical attacks to create a blackout [339].

A reliable smart grid requires layered protection applications that consist of a cybernetic infrastructure that limits adversary access and limits the operation of the transmission accurately during an attack.

### 4.1.2. Cluster 2 (Green): Authentication in Smart Cities

One of the mechanisms for protecting data and information is access control policies for systems. Access control helps to prevent unauthorized people from entering the virtual and/or physical environment and engaging in unauthorized behavior. By ensuring access control, the integrity of employees and service providers is provided, as well as the integrity of data and information [337].

Over the years, the growing number of companies that select an outsourcing strategy for managing the entire IT infrastructure has been noticed. This interest is often motivated by the high investment in current IT security solutions, which require constant adaptations to the environment [340]. On the other hand, this need for adjustments makes many outsourced companies assume that their technology service providers are responsible for data control. However, when it comes to information security and compliance, the organization promoting the leading service remains responsible for all the information it has, especially if the company wants to obtain more profitable results from the data.

In this context, the objective of managers is to ensure that the large volumes of data collected and stored by their organizations can be used as instruments that help to generate better business strategies, making companies more objective and eliminating any types of confusion that may be caused by the total amount of information to be evaluated, adopting control systems with different types of possibilities, which can be physical or digital [219,341].

### 4.1.3. Cluster 3 (Blue): Cyberattacks in Smart Cities

The popularization of cloud computing encouraged the development of new businesses and reduced the need for high investment in IT infrastructure for small businesses, in particular. On the other hand, cybersecurity has become a significant concern for these companies. In the virtual environment, attackers create different threats to the systems of different businesses, from financial services agencies to sizeable industrial control systems [252,342,343]. Attack methods vary widely, using simple techniques to exploit the vulnerabilities of access and communication protocols or through combined operations for the use of multiple web bots [344].

One of the strategies to combat these threats is intrusion detection, the most effective security mechanism for detecting internal attacks that consists of the process of monitoring and analyzing events that occur in a computer system or network in search of patterns of possible security incidents. For the authors, these security incidents are violations or threats to security policies defined as attempts to compromise the reliability, integrity, or availability of system resources [345–348]. Many types of malware can be programmed to destabilize the operation of a system, such as viruses, worms, Trojans, and backdoors [349,350].

One of the main concerns of the authors is that the automatic detection of known and unknown kernel rootkits on virtual machines is becoming an urgent problem. For the virtual environment, an Intrusion Prevention System (IPS) is considered an extension of the Intrusion Detection System (IDS) and can be executed when threats or malicious activities are detected [351]. Thus, there is a tendency for new solutions to be made available to promote a kind of digital investigation and detect cybercrimes [352].

### 4.1.4. Cluster 4 (Yellow): Security Platforms for Smart Cities

For current businesses, one of the main assets is useful information. However, defining the monetary value of threats to this information can be a complex process. Economic decision models have been used to quantify the cyberattack process or demonstrate the intruder's detailed behaviors [353,354]. The advances in this area are mainly based on structured ways to present the consequences of the inventions to the IT Manager and recommend viable actions to avoid possible theft of information, for example, which represent the highest external cost, followed by the costs associated with interrupting operations of business [355].

To deal with rapidly evolving threats and risks, different approaches can be used to perform the command injection attack on the cyber component in the SCADA system: Model of the SQL Injection Attack, Model of the Secure Sockets Layer (SSL), Model of the Address Resolution Protocol, Model of the Buffer Overflow Attack [64,356]. In this context, dealing with an analytical decision model under conditions of uncertainty can be important for IT managers when planning information security programs.

### 4.1.5. Cluster 5 (Pink): Evaluation of Threats to Cybersecurity

The domain of cybersecurity threats is directly related to discussions about cybersecurity control and data in online services. Form IT advancement, new communication technologies, and control methods may allow better regulation of the smart grid; however, they also introduce serious threats to cybersecurity. In the Digital Age, security is the keyword. For the authors, having reliable data, systems, and people is indisputable because cyberattacks happen frequently, and systems capable of preceding an attack are essential [357].

Cyberattacks may also cause cascading failures in a power system, thus posing a serious threat to national infrastructure. Because of this, the authors suggest that the preconditions for managing cybersecurity risks are discovering incidents, collecting data, and viewing that data [174,358]. Three principles support this management cycle: maintaining the right data, robust IT infrastructure (systems), and an appropriate scope of sharing (people).

Impact analysis of threats is necessary to analyze the consequences of interruptions in the flow to protect and enable the evolution of business through technology, as well as to monitor users, observe the behavior, and monitor the development of attacks. Therefore, making potential threats clear can improve the protection shield and allow for new business opportunities [341].

The idea of resilience against a cyberattack, in addition to helping to know how to deal with a situation for which companies are not prepared, is to recognize the complexity of a scenario and have a contingency plan and defenses at different levels of security. In this way, it is possible to mitigate possible impacts resulting from cyberattacks [359].

In this sense, performing defensive security planning is essential, as the systems will cease to function over time, generating large potential losses for companies. Hong et al. [360] comment that investing in business cybersecurity is essential, given that criminals focus on operating systems with security gaps that have not been fixed or that have not yet been updated to a newer version. This vulnerability increases the risk and highlights the importance of investing in a consistent monitoring process [361].

In several countries, cyber defense constitutes a national security framework in which states establish policies at all levels (public and private) to guarantee individual freedoms and to respond to aggressions and invasions by developing response and cooperation systems [362]. Taking these security policies as a reference related to cyber resilience, emerging countries can adopt the definition of tasks and missions to establish security standards in the public and private environment, highlighting the specific criticality of the IT infrastructure [363].

### 4.1.6. Cluster 6 (Purple): Cybersecurity and Society

One of the most recent discussions related to cybersecurity has involved the influence of social aspects applied to the advancement of IT solutions [364]. Given the increase in urbanization around the world, growing populations are overloading the social services provided by the government, which in turn aims to facilitate the processes that citizens trust and need. This aspect motivates the emergence of the concept related to the construction of functional cities, which allow residents to have happier and healthier lives in a smart environment. In so-called "cities of the future," communities and organizations make extensive use of information technology to ensure broad and efficient access to early childhood education programs, professional recycling, and other vital social and citizenship programs that can be digitally connected [365].

However, one of the central points of the discussion is that there is no human consensus on ethics, especially on the sharing of information and space. Ethics is interpreted as a concept applied to a given context and, therefore, extremely complex to be programmed [366]. For the authors, machines need to be programmed with the minimum ethics necessary to avoid consequences in the future, but when human ethics is assumed, it does not seem to be the best model for teaching machines [367]. This motivation stimulates the discussion

about new ethics, something close to the consensus that would be used to program the artificial intelligence of the future.

This cluster involves the relationship between cybersecurity incidents and understanding of human behavior, in particular, incidents registered in business environments. For the authors, the protection of confidential data in companies is fundamental for business development and allows risks to be minimized [366]. This protection is based on two factors: technical and human factors. In general, the functional element involves investing in IT solutions that ensure access control mechanisms, user identification, antivirus systems, and restricted access to components of the IT infrastructure. On the other hand, the human factor refers to the user's perception of information security related to the knowledge of vulnerabilities and severity of risk regarding the lack of corruption of data and information, information shared on the internet, practices, and experiences with information security in the business environment.

The relationship between these factors raises a relevant discussion for the development of protection strategies that ensure control over the influence of human behavior in detriment to the investment of technical factors [342]. Cybersecurity strategies can be developed based on the perception of human behavior in an integrated manner with specialized solutions and IT governance to monitor the movement of confidential data that can be transmitted outside the company. The destructive consequences of spills are clear, but the risks caused by the human factor are often overlooked and can cause a company to go bankrupt. A situation that can exemplify this loss is when a sales employee improperly uses customer data, being able to use private information regarding business transactions in an unworthy manner [366].

In this context, awareness must be an ongoing effort to educate employees about policies, threats to data and information security, and how to deal with them [368]. Protection Motivation Theory can be applied to understand and develop a culture that motivates employees to maintain safe practices in their daily lives and transform awareness training into something personal. In addition to these theories, educational games can help support the concepts of awareness and improve understanding of possible incidents and their impacts on the organization and its business [128].

### 4.2. Top Authors with the Highest Number of Citations

Table 3 presents the 20 highly cited articles in information security and smart cities in the Scopus database.

**Table 3.** The 20 most cited articles on information security and smart cities.

| Index | Author | Total of Citations | Title | Reference |
|-------|--------|--------------------|-------|-----------|
| 1 | Farahani et al., 2018 | 1001 | Towards fog-driven IoT eHealth: Promises and challenges of IoT in medicine and healthcare | [155] |
| 2 | Rathore et al., 2016 | 996 | Urban planning and building smart cities based on the Internet of Things using Big Data analytics | [54] |
| 3 | Dagher et al., 2018 | 746 | Ancile: Privacy-preserving framework for access control and interoperability of electronic health records using blockchain technology | [101] |
| 4 | Biswas et al., 2016 | 746 | Securing smart cities Using Blockchain Technology | [369] |
| 5 | Elmaghraby et al., 2014 | 640 | Cyber security challenges in smart cities: Safety, security and privacy | [15] |

**Table 3.** *Cont.*

| Index | Author | Total of Citations | Title | Reference |
|---|---|---|---|---|
| 6 | Xie et al., 2019 | 630 | A Survey of Blockchain Technology Applied to smart cities: Research Issues and Challenges | [252] |
| 7 | Zhang et al., 2017 | 620 | Security and Privacy in smart city Applications: Challenges and Solutions | [370] |
| 8 | Sivanathan et al., 2019 | 579 | Classifying IoT Devices in Smart Environments Using Network Traffic Characteristics | [371] |
| 9 | Sharma et al., 2017 | 500 | Block-VN: A Distributed Blockchain-Based Vehicular Network Architecture in smart city | [372] |
| 10 | Khatoun et al., 2016 | 473 | Smart cities: concepts, architectures, research opportunities | [373] |
| 11 | Djahel et al., 2015 | 436 | A Communications-Oriented Perspective on Traffic Management Systems for Smart cities: Challenges and Innovative Approaches | [374] |
| 12 | Singh et al., 2020 | 429 | Block IoT Intelligence: A Blockchain-enabled Intelligent IoT Architecture with Artificial Intelligence | [242] |
| 13 | Sharma et al., 2018 | 411 | Blockchain-based hybrid network architecture for the smart city | [375] |
| 14 | Angelidou et al., 2017 | 390 | The Role of smart city Characteristics in the Plans of Fifteen Cities | [376] |
| 15 | Rathore et al., 2018 | 330 | Exploiting IoT and big data analytics: Defining Smart Digital City using real-time urban data | [377] |
| 16 | Memos et al., 2018 | 352 | An Efficient Algorithm for Media-based Surveillance System (EAMSuS) in IoT smart city Framework | [188] |
| 17 | Aloqaily et al., 2019 | 353 | An intrusion detection system for connected vehicles in smart cities | [56] |
| 18 | Braun et al., 2018 | 307 | Security and privacy challenges in smart cities | [7] |
| 19 | Esposito et al., 2021 | 297 | Blockchain-based authentication and authorization for smart city applications | [225] |
| 20 | Qiu et al., 2017 | 215 | Heterogeneous ad hoc networks: Architectures, advances and challenges | [378] |

These results show the importance and impact of smart city studies. Another important fact is that in recent years, new challenges regarding application information security in smart cities have emerged due to new technologies. As an output of the analytical process, papers have addressed these new issues and consequently have a high potential for being more cited in the future. For instance, the automation of vehicles in the field of intelligent transport systems [379] and human beings as potential targets for cyberattacks or even participating in a cyberattack with ethical implications for society.

### 4.3. Most Active and Cited Journals

Journals play an essential role in the development of a research area. Table 4 reports the most prominent journals in the number of publications on cybersecurity in the Scopus database and their impact factor in 2022.

**Table 4.** Journals and Impact Factors for information security and smart cities related literature.

| Subject Areas | Source | Impact Factor 2022 | # of Article |
|---|---|---|---|
| Computer Science | Computers and Security | 5.6 | 262 |
| | Future Generation Computer Systems | 7.5 | 712 |
| | IEEE Access | 3.9 | 139 |
| | IET Information Security | 1.4 | 23 |
| | Computer Communications | 6 | 323 |
| | IEEE Security and Privacy | 1.9 | 54 |
| | Computers in Human Behavior | 9.9 | 60 |
| | Information Technology and People | 4.4 | 63 |
| | International Journal of Communication Systems | 2.1 | 256 |
| | International Journal of Software Engineering and Knowledge Engineering | 0.9 | 12 |
| Social Sciences | Computer Law and Security Review | 2.9 | 164 |
| | Technological Forecasting and Social Change | 12 | 346 |
| | Public Administration Review | 8.3 | 13 |
| | Technology in Society | 9.2 | 145 |
| | Journal of Intellectual Capital | 6 | 64 |
| | Behavior and Information Technology | 3.7 | 88 |
| | International Journal of Human Computer Studies | 5.4 | 27 |
| | Business Horizons | 7.4 | 58 |
| | International Journal of Accounting Information Systems | 4.6 | 12 |
| Business, Management and Accounting | International Journal of Information Management | 21 | 130 |
| | Government Information Quarterly | 7.8 | 157 |
| | Information Technology for Development | 4.261 | 47 |
| | European Journal of Operational Research | 6.363 | 33 |
| | Information Sciences | 8.1 | 131 |
| Energy | Energies | 3.2 | 195 |
| | Sustainability | 3.9 | 76 |
| | Energy Research and Social Science | 6.7 | 151 |
| | Journal of Cleaner Production | 11.1 | 465 |

It is worth mentioning that the top journals showing that the topic of information security and smart cities has attracted the attention of researchers from different fields. Because smart city is a multidisciplinary field, scholars often struggle to figure out the most appropriate outlet for their research that would have a significant impact. The information reported in this table indicates this willingness to publish in each specific area.

### 4.4. Country Co-Citation Analysis

In the next phase, the collaboration networks among countries were highlighted, as presented in Figure 4. The figure shows the distribution of countries with the most co-authorships. The clusters are indicated by circles and colors, explaining the proximity of the countries and the associations between co-authorships, while the edges illustrate how researchers' production is expanding. Notably, China (*n* = 462) presents the bigger production, followed by India (*n* = 411), the United States (*n* = 239), the United Kingdom (*n* = 146), Saudi Arabia (*n* = 125), South Korea (*n* = 102), Pakistan (*n* = 93), Australia (*n* = 71), Italy (*n* = 65), Spain (*n* = 63), Canada (*n* = 54), Taiwan (*n* = 51), Brazil (*n* = 49), Malaysia (*n* = 46), Turkey (*n* = 45), United Arab Emirate (*n* = 38), and Iran (*n* = 35).

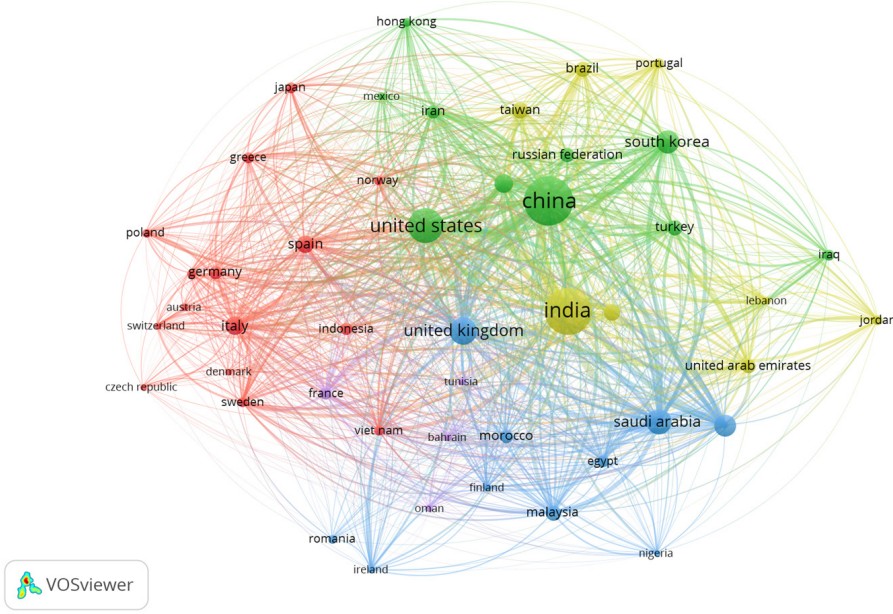

**Figure 4.** Collaboration networks on information security and smart cities among countries.

As can be seen, the research collaborations appear with a higher level of intensity among countries of the European Union and those of North America. In addition, there is also a collaboration network among Asia, North America, and Europe. Research collaboration in cybersecurity indicates the complexity of the interrelations and the opportunity for future cooperation. Also, the results allow three inferences to be drawn: countries with the most cooperation may offer practical implications for society through the partner with industries; academic experts affiliated with these countries can provide knowledge as references on the issue; and the contributions developed by the authors can serve as guidelines for other researches.

### 4.5. Keyword Co-Occurrence Analysis

Figure 5 highlights the network visualization for the most common terms used in the authors' keywords. The network reports the most relevant keywords of these items in terms of occurrences and their interactions between documents. A total of 267 keywords emerged, with at least one occurrence [380,381]. From this network, 36 items are considered independent, in which case the item does not bring any significant contribution to designing applicable queries and identifying pertinent empirical surveys. As expected, "Smart

city," "network security," and "security systems" stand out as the most common terms. However, upon closer examination of the other circles, correlations emerge concerning the topics presented in the paper. The blue cluster is associated with the analysis of surveillance and the application of methods for recognition. The green cluster is related to the analysis involving smart transportation. The purple cluster provides limited information about "cryptography" and contract management. The red cluster focuses on urban livability and its interaction with technology. The remaining terms do not exhibit significant expressiveness to readiness.

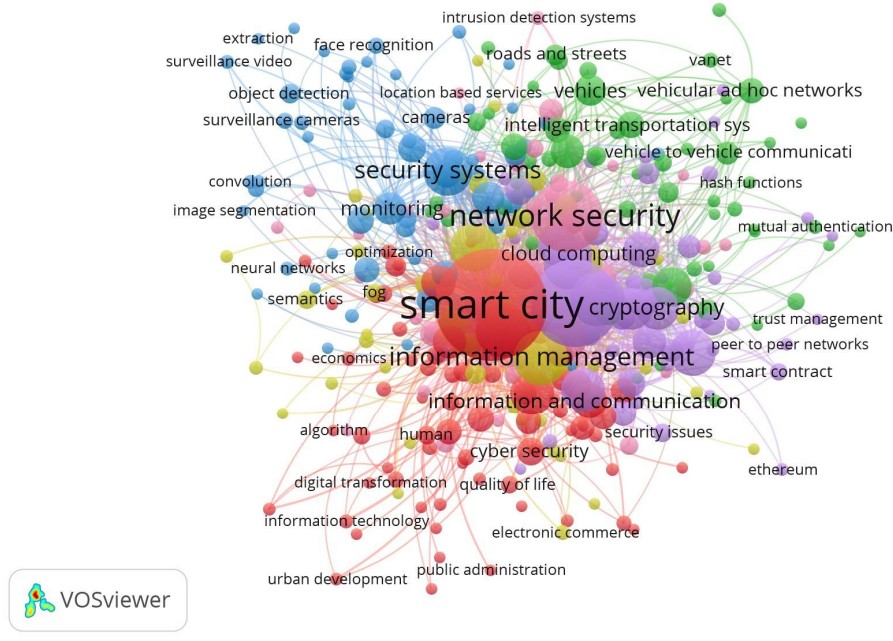

**Figure 5.** Most Relevant Keywords.

Table 5 details the keywords with the highest occurrences and interactions based on the set of complete articles published in journals indexed in Scopus, ranging from 2015 to 2023, as based on the network shown in Figure 5.

**Table 5.** High-frequency keywords for searches in the area between 2015 and 2023.

| High-Frequency Keywords | Occurrences |
| --- | --- |
| Smart city | 1146 |
| Internet of Things | 699 |
| Network Security | 470 |
| Security | 374 |
| Computer Security | 324 |
| Cyber–Physical System | 314 |
| Data Information | 291 |
| Blockchain | 198 |
| Energy Efficiency | 174 |
| Energy Security | 166 |
| Cryptography | 156 |
| Green Computing | 141 |
| Information Security | 139 |
| Smart Grid | 133 |
| Sustainable Cities | 131 |
| Urban Development | 127 |
| Urban Planning | 123 |
| Accident Prevention, Attack Detection | 119 |
| Authentication, Authentication Protocols | 117 |
| Intelligent Transportation Systems, Information Exchanges | 116 |

**Table 5.** *Cont.*

| High-Frequency Keywords | Occurrences |
|---|---|
| Privacy Preservation | 115 |
| Public Key Cryptography | 110 |
| Network Protocols, Security Vulnerabilities | 102 |

These results demonstrate that among the articles published, the keywords smart city and internet of things have the highest occurrence rates, which demonstrates the growing interest of researchers in topics related to information security and smart cities.

*4.6. Methods in Cybersecurity*

Methods play an essential role in the development of a research area. We have included Table 6 with 11 main cybersecurity methods applied in main areas such as Computer Science, Engineering, Mathematics, Social Sciences, Business Management, and Accounting.

**Table 6.** Cybersecurity methods and applications according to main areas.

| Method | Computer Science | Engineering | Mathematics | Social Sciences | Business, Management and Accounting | Total |
|---|---|---|---|---|---|---|
| Risk Management | 57 | 32 | - | 19 | 21 | 129 |
| Machine Learning | 48 | 17 | 7 | 9 | 11 | 101 |
| Game Theory | 28 | 17 | 9 | 8 | 2 | 64 |
| Neural Network | 17 | 15 | 4 | - | 5 | 41 |
| Data Mining | 25 | 5 | 2 | - | 5 | 37 |
| Deep-Learning | 18 | 7 | 3 | 1 | 2 | 33 |
| Blockchain | 17 | 8 | 3 | 2 | 3 | 33 |
| Fuzzy Theory | 16 | 6 | 5 | - | 2 | 29 |
| Bayesian game | 6 | 3 | 2 | 2 | 2 | 15 |
| Software-Defined Networking | 6 | 2 | 2 | - | 1 | 11 |
| Natural Language Processing | 4 | 2 | - | - | 1 | 7 |

These results demonstrate that Management Risk and Machine Learning have a total of 129 and 101 articles published, respectively. They allow the consideration of important factors that can lead to better decision-making in information security, and smart cities have become more widely used in actions focused on defense strategies.

**5. Discussion**

The discussion on information security and smart cities is not restricted to the area of computer science. The concern about data and information security is multidisciplinary and influences the evolution of different types of business. Health professionals, government institutions, academic environments, and several other stakeholders benefit from the opportunities for advancing research while they can take advantage of this study to indicate potential solutions and improve the level of information security, predicting the consequences of information loss [328,331]. For this, when planning on cybersecurity, it is necessary to prioritize strategic processes, actions, and tools that will be implemented or used, both for the organization, for the government/public administration, and for society in smart cities [376].

Smart cities use information and communication technologies to improve the quality of life of their inhabitants, making public services more efficient and creating innovative solutions to urban challenges [15]. However, as cities become more connected and dependent on technology systems, information security becomes an ever-increasing concern. Citizens' data, as well as operational information on critical city systems, can be at risk

from cyberattacks. Therefore, smart cities must have a comprehensive information security strategy to protect their systems and data [370]. This involves implementing cybersecurity measures at all layers of the city's infrastructure, from the communication network to IoT (Internet of Things) devices and data management systems [54,370].

To decrease the probability of a cyber threat causing damage, some cyber security measures should be implemented, such as Encryption, Authentication of users, Network Security, Cyber security training, and Regular software updates [90,382,383]. These shared vulnerabilities can be exploited by hackers and other malicious users to compromise city security, directly affecting citizens' lives. For example, a cyberattack on a traffic management system can lead to severe congestion and delays in emergency services. Some of the most common shared vulnerabilities in smart cities are weak passwords, delayed software updates, and unsafe IoT devices. This work contributes to presenting new information security technologies to minimize shared vulnerabilities in smart cities; it is essential to adopt comprehensive cybersecurity measures.

A challenge for developing countries will be the integration of smart cities. The decision to plan information security for the management of cities is essential to guarantee engagement in municipal services through intelligent digital systems. So, the smart city ecosystem requires new skills and competencies in various ways through strategic partnerships and contracts with service providers [373]. Maintaining a safe and smart city involves creating a public/private infrastructure to carry out activities and provide technologies that protect and protect citizens' information [286].

Four main considerations should be address regarding smart cities security:

1.  Strategies for artificial intelligence and shared communications are necessary, ensuring opportune analysis of data/information flow through smart cities systems to detect threads and ensure the secure delivery of what must be communicated from one end to the other [22,384], and consequently providing the necessary confidentiality and privacy in communications [385];
2.  Physical and cyber threats come from many areas, including state-sponsored critical infrastructure, criminals, natural disasters, and neglect of human agents [307,386,387], all opening several security holes that must be foreseen in risk containment plans to guarantee the integrity of the information that passes between the systems involved, demanding a smart cybersecurity architecture that can cover these risks [292];
3.  Integrated operational management activities and knowledge sharing to prevent, mitigate, respond, and recover from incidents [388].
4.  Acquiring emerging technologies that facilitate risk assessment ensures appropriate physical security and cybersecurity measures [172].

### 5.1. Addressing the Research Questions

The literature review developed had three research questions as its core, as presented in the methodological section. Based on literary findings, directions on these questions will now be presented.

RQ1—What are the patterns of information security applications found in research on smart cities?

This question can be addressed with the six clusters presented in Table 2, separating each cluster according to the main application domain areas, as follows:

(a)  Smart Grids and Power Supply: this cluster covers works that mention applications that can cover information and cybersecurity on smart grids as a component of smart city systems to ensure efficient, safe, and sustainable power supply for citizens [226]. Smart grids cover topics such as bulk generation, transmission, distribution, customers, markets, service providers, and operations [78].
(b)  Authentication as a security mechanism: this cluster covers applications regarding the control access policies and strategies for data protection in smart city systems, especially considering the large data volumes that are inherent to these systems [291].

Authentication mechanisms are projected to ensure privacy, trust, and reliability in the information and communication flows [51] to protect against invasion by attackers masquerading as legitimate users of the system [85].

(c) Cyberattack prevention/detection in smart cities: this cluster focuses on strategies to prevent or detect cyberattacks or vulnerabilities that may facilitate these attacks in the smart cities context, observing the best practices and methods to be applied in protecting involved systems [280]. The lack of these strategies can cause, for instance, theft of a user's sensitive data, utility fraud, and grid instability [1]. In other words, this can be considered a cluster containing works presenting core concepts and tools that are transversal to all other clusters.

(d) Security platforms for smart cities: this cluster involves not only technological platforms but the whole organizational and business instances needed to promote security to smart cities-related services and systems [60]. The main idea is to deliver quality of life for the users of these services and systems, which are any citizen in a smart city area [302]. Quick and efficient managerial decision-making is the main concept to ensure security platforms operate successfully in preventing risks from becoming events negatively affecting smart city services delivery for citizens [302]. These platforms are a means for aggregating concepts of the other five clusters, as can be understood by the diagram in Figure 6 in the answer for RQ2, synthesizing the relationships between all clusters of applications.

(e) Evaluation of threats to cybersecurity: this cluster deals with ways to evaluate threats to the smart cities systems, facilitating, for example, the design and management of security platforms and ensuring the necessary indicators and related analysis to promote the detection and prevention of cyberattacks [311,319]. It covers from devices to threat evaluation techniques, which can be used in support of security measures planning [6,54].

(f) Cybersecurity and society: this is the most comprehensive cluster, involving all the elements needed to promote cybersecurity for society, considering smart cities as cyber–physical systems [328]. It covers legal and ethical concepts, passing by managerial strategies and reaching the technical level with the frameworks of techniques/tools to ensure cybersecurity for people [333].

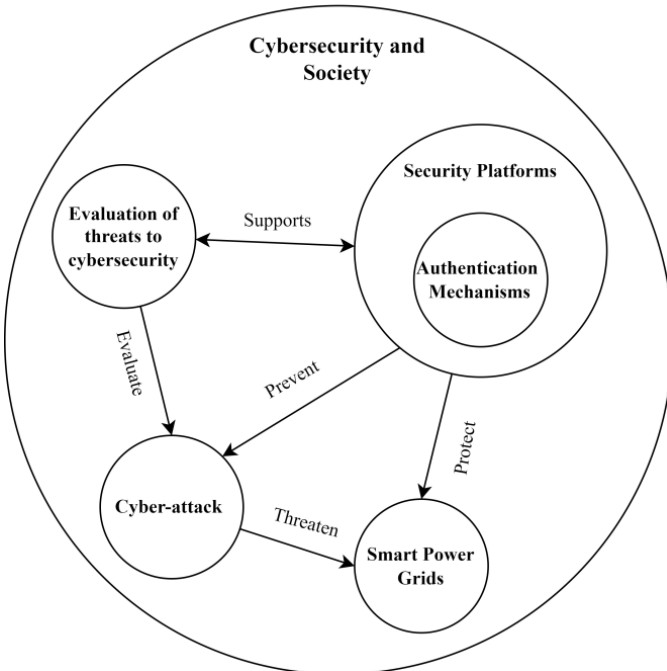

**Figure 6.** Relationships between the concepts involved in the applications clusters.

RQ2—What are the most demanding areas for information security in smart cities studies?

To answer this question, Figure 6 was created seeking to highlight the dynamics between the previously observed clusters. It should be noted that the diagram in the figure does not present a composition of works/authors found in the literature as in Figure 3, but the conceptual alignment and flow between the clusters.

Through this figure, we can see that the evaluation of threats to cybersecurity and security platforms are great "providers" within the set since several sectors within smart cities require constant monitoring and adequate analysis to detect threats, such as cyberattacks, and these assessments are fundamental to support the structuring and operationalization of security platforms. In turn, security platforms constitute essential components in smart cities to guarantee the dynamics of security in related systems, including smart grids, providing means for continuous evaluation of threats and preventing any kind of unauthorized access.

Another address that can be given to RQ2, as detected in the literature, is when it comes to the service provision sector. The most notable is energy supply, which gained prominence in a cluster that contained the largest number of jobs compared to the other clusters. However, other sectors receive several mentions in the literature, with the healthcare area being one of the most prominent. Table 3 indicates, for example, the work by Farahani et al. [155] in the line of IoT in medicine and healthcare as the one with the highest number of citations within the bibliographic base built for the bibliometric review. The third most cited work, by Rathore et al. [101], is also related to healthcare, proposing a framework based on blockchain for electronic health records. By the way, several of the works among the most cited are about the blockchain and related technologies appear in seven works (see [101,225,242,252,369,372,375]).

Blockchain, as a set of technologies for databases to ensure transparent data sharing, can be considered a core concept for the project of security platforms and systems in smart cities, being a transversal technical area that can be considered for smart grids and healthcare information security. Other areas, such as urban planning and building [54], transport/vehicles, and traffic control systems [56,372,374], can also be mentioned here as highlighted, as they are critical for the proper operation of smart cities, delivering quality of life and effective services to citizens.

RQ3—What research has the most influence on the application of information security and smart cities?

This question is also easily answered by the list of works in Table 3. It is intertwined with the comments made in the last two paragraphs of the previous section dedicated to RQ2. Following, the objectives of the top five most cited works are presented.

Farahani et al. [155], with 1001 citations, presented a survey of IoT Health and put forth a holistic eHealth ecosystem that encompasses various layers, including mobile health, assisted living, e-medicine, implants, early warning systems, and population monitoring.

Rathore et al. [54], with 996 citations, presented the proposal of a complete smart city architecture, also considering urban panning with data analysis on Big Data based on IoT.

Dagher et al. [101], with 746 citations, presented the proposal of a blockchain-powered framework designed to enable secure, seamless, and efficient access to medical records for patients, healthcare providers, and third parties while maintaining the privacy of sensitive patient information.

Biswas et al. [369], also with 746 citations, introduced a security framework that combines blockchain technology with smart devices, creating a secure communication platform within a smart city.

Elmaghraby et al. [15], with 640 citations, presented a survey on cybersecurity challenges, exploring two interconnected challenges, namely security and privacy. Additionally, they introduced a model for the interactions among individuals, servers, and IoT devices as the key elements in a smart city, emphasizing the necessity to safeguard these interactions.

*5.2. Theoretical and Practical Implications*

The results contribute to developing a practical perspective in computer science, particularly providing a conceptual framework integrated with information security and smart cities knowledge and leading research in the world. IT security professionals can take advantage of this study by using this structure as a reference to design new solutions in cybersecurity and formulate specific security policies to combat and prevent cyberattacks in smart cities. Moreover, this study shows the importance of developing information security strategies with a focus on user behavior in the city, characterized as the primary agent that causes security failures in IT solutions. In addition, IT researchers can obtain guidance to explore new fields of research, develop new trends and perspectives, develop applications to fill gaps in the literature and provide attention to different types of problems in information security and smart cities, which highlights the validity and relevance of this work.

Clustering bibliometric networks through co-citation analysis has practical contributions to the business area. By integrating knowledge between the disciplines of information and computing systems, managers and practitioners can quickly identify the most relevant concepts and best practices concerning information security and smart cities and perception of human behavior, smart power grid, online services, prevention systems for cyberattacks, the critical cyber infrastructures, threats, resilience, and social prospects of cybersecurity, designed by the clusters of co-citation analysis. As stated by [389], such a repository of terms associated with the scientific literature is a strategic tool for the continuous improvement of business, which can designate appropriate software features or necessary maintenance for the security of information systems and support decision methods in the treatment and prevention of information security incidents. This systematic view can also highlight organizations' responsibility of managers for smart city decisions related to control and data privacy and potential correlations between data security and the organization's value judgments on security devices.

Although developments and research related to the creation of control software, infrastructure improvement, risk prevention, and failure prevention, investment in IoT and Data solutions Science have increased in the last ten years, as shown by the results of this research, cybersecurity is still treated as a secondary element in government organizations and institutions in developing countries. In this context, the acquisition of new IT solutions must be considered a strategy as important as the investment in cybersecurity, as it can directly affect the users' perception of smart cities. Service providers must adhere to service-level agreements regarding system operation, data generation, and the use and sharing of information. Additionally, they should undergo privacy impact assessments to ensure compliance with privacy regulations and protect individuals' personal information. By enforcing these requirements, organizations can ensure that service providers maintain a high standard of service delivery, respect privacy rights, and safeguard sensitive data.

This research presents an integrative theoretical framework conceptualized in the presentation of the state of the art on the scope of application and development of the term "information security and smart cities." The theoretical framework presented can provide conceptual support to researchers and professionals in the field and can be used as a reference for understanding the connections between the lines of research, the composition of clusters of researchers, and the relationship between related areas, and can serve as a conceptual basis for the cybersecurity planning project in different businesses.

## 6. Conclusions

This study reported the construction of a systematic review involving bibliometric aspects, oriented to the identification of the main applications of the information security and smart cities concept, such as cybersecurity and human perception behavior, cybersecurity and smart electrical network, cybersecurity control and data in services online and intrusion detection for cybersecurity. The analysis, spanning articles from 2015 to 2023 in Scopus-indexed journals, leveraged VOSviewer software for mapping global researchers and their



contributions. The findings underscored the interdisciplinary nature of information security and smart cities, emphasizing their relevance beyond computational sciences.

The study's outcomes offer valuable insights for managers, professionals, and academics across diverse domains, highlighting opportunities for exploration within the literature of cybersecurity in smart cities. The implications of information security and smart cities extend beyond computational sciences, influencing business actions, social development, and service enhancement. The results emphasize the need for interdisciplinary approaches in cybersecurity research, indicating collaboration across engineering, administration, psychology, economics, and law. Furthermore, the study advocates for a holistic perspective in cybersecurity research, promoting interdisciplinarity and encompassing ethical considerations for effective business strategies in the digital era.

Noteworthy findings include the identification of leading countries in cybersecurity studies, with China, India, the United States, and the United Kingdom taking the forefront. The study observes a lack of exploration in cybersecurity studies in developing nations, often attributed to technological limitations. It also notes a growing trend of international collaboration among researchers in the field. There is a need for research in cybersecurity solutions, particularly in the context of virtual service systems such as telehealth services within smart cities.

Although the work has a full scope in information security and smart cities, some limitations can be mentioned, such as potential oversight of frontier applications during the detailed analysis and the lack of considerations for cybersecurity software in this review. This could be an exciting gap for future research, including a comprehensive assessment of cybersecurity software options similar to the work of Daraio et al. [389] on efficiency frontier applications.

There is a growing call for collective initiatives and educational campaigns centered on information security. A deeper public understanding in this domain can catalyze a stronger trust in the technologies underpinning smart cities, bolstering their adoption and seamless integration into citizens' daily lives. Information security is undeniably a foundational pillar for the successful assimilation of these technologies. Consequently, it becomes imperative to address not only the technical facets but also the subjective and objective dimensions highlighted in this study, which impact the global landscape.

**Author Contributions:** Conceptualization, T.P. and T.C.C.N.; methodology, T.P., L.C.B.d.O.F., R.C.P.d.O. and T.C.C.N.; software, T.P.; validation, V.D.H.d.c., T.C.C.N. and C.J.J.F.; formal analysis, T.P. and L.C.B.d.O.F.; investigation, T.P., R.C.P.d.O., T.C.C.N. and V.D.H.d.C.; resources, T.P.; data curation, T.P.; writing—original draft preparation, T.P. and V.D.H.d.C.; writing—review and editing, V.D.H.d.C. and C.J.J.F.; visualization, T.P.; supervision, T.C.C.N. and V.D.H.d.C.; project administration, T.P.; funding acquisition, T.P. All authors have read and agreed to the published version of the manuscript.

**Funding:** This research received no external funding.

**Data Availability Statement:** The data presented in this study is only contained in the article itself.

**Acknowledgments:** We want to acknowledge the support from the Coordenação de Aperfeiçoamento de Pessoal de Nível Superior (CAPES, Brazil), the Conselho Nacional de Desenvolvimento Científico e Tecnológico (CNPq, Brazil), the Universidade Federal do Pará (UFPA, Brazil), the Universidade Federal de Alagoas (UFAL, Brazil), the Universidade Federal de Pernambuco (UFPE, Brazil), and the Universidade Federal Rural do Semi-Árido (UFERSA, Brazil).

**Conflicts of Interest:** The authors declare no conflict of interest.

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
