# Peer review of "Information Security Applications in Smart Cities: A Bibliometric Analysis of Emerging Research"

_futureinternet, doi:10.3390/fi15120393_

Round 1
Reviewer 1 Report
Comments and Suggestions for Authors
Dear Authors,
The paper addresses an important and current topic of information security applications in smart cities. However, its content and structure require minor modification and enrichment:
Please explain why did you choose to analyse the publications in the period of 2015 to 2023, if the year 2023 has not finished yet?
Figure 3 – please explain what does the size of circles mean?
Figure 4 and 5 should be discussed more deeply by adding the explanation and discussion of each cluster. I suggest adding the information about the meaning of: colours of circles, size of a caption (or circles) and the distance between terms.
Discussion section should be much more developed, presenting: a) the main findings from the study enriched with cited articles; b) answers to research questions mentioned in section 3 Materials and Methods.
In section 4.5. Keyword co-occurrence analysis, the period of 2014-2022 is mentioned, while in Materials and Method the period of 2015-2023. It should be corrected.
In Table 3. The 20 most cited papers, the year should be added to the name of author.
Line 452-472, I suggest to include the frequency keywords in the table to make it more clear.
Tables and Figures should be formatted according to journal requirements, especially italics and width.
Figure 2 – the form of graph should be changed, or the title of graph, as it shows the status of number of publications per year. It should be presented as a bar chart with tendency line or the title of graph should indicate that it presents the trend.
Title of section 4.2 “Top authors with the highest number of publications and citations” should be changed as it shows only the numbers of citations.
The quality of Figure 4 should be better and more visible.
Comments on the Quality of English LanguageMinor editing of English language required
Author Response
Dear reviewer,
We thank you for your time and attention to our manuscript. We are certain that your comments and recommendations ensured better detailing of our text, helping us to add or improve what was already reported.
Below we present our responses, point by point, to their comments and recommendations.
Please, see the response letter we attached to indicated what was done to address your recommendation.
Also, see the PDF of the manuscript we sent the blue highlights indicating all the textual areas or other elements (figures and tables) we changed or included in this new version.
With our regards,
The authors

Reviewer 2 Report
Comments and Suggestions for Authors
The present study reported the construction of a systematic review, involving bibliometric aspects, oriented to the identification of the main applications of the information security and smarty city concept, The manuscript is interesting, novelty and well-written. I suggest publishing it in its present form.
Author Response
Dear reviewer,
We express our gratitude for your dedicated time and attention to our manuscript. We are truly thankful for the valuable feedback on our research reported in this manuscript.
With our regards,
The authors
Reviewer 3 Report
Comments and Suggestions for Authors
In this paper, authors surveyed Information Security Applications in Smart Cities. The topic is interesting and also useful for the readers. However, to improve the quality of the manuscript. My comments are given below.
1) The introduction section is short. Please extend by adding motivation, benefits of this research and also your contribution in bullets.
2) Please add a few state of the art models along with pros and cons.
3) Please highlight your research question.
4) Related work section is missing in this paper. Please add this section and also mention pros and cons in detail.
5) The quality of figures should be improved in the revised version. Currently, the quality of figures is not good.
6) Please add more recent references in the references section using top journals.
7) The conclusion section is very long. It should be shortened and supported by future work.
8) Please add more comparison tables and also mention pros and cons in a tabular form.
Comments on the Quality of English Language
The english should be checked by grammerly and also check for the grammer mistakes.
Author Response
Dear reviewer,
We thank you for your time and attention to our manuscript. We deeply appreciate your insightful feedback, whose input greatly enhanced the quality of this work.
Below we present our responses, point by point, to their comments and recommendations.
Please, see the response letter we attached to indicated what was done to address your recommendation. Also, see the PDF of the manuscript we sent the blue highlights indicating all the textual areas or other elements (figures and tables) we changed or included in this new version.
With our regards,
The authors

Round 2
Reviewer 3 Report
Comments and Suggestions for Authors
The authors has incorporated all necessary changes in the mnauscript.
Comments on the Quality of English LanguageIt is fine.